# Selection of XAI Methods Matters: Evaluation of Feature Attribution Methods for Oculomotoric Biometric Identification

**Daniel Krakowczyk**\*                    DANIEL.KRAKOWCZYK@UNI-POTSDAM.DE
*Department of Computer Science, University of Potsdam, 14476 Potsdam, Germany*

**David R. Reich**                         DAVID.REICH@UNI-POTSDAM.DE
*Department of Computer Science, University of Potsdam, 14476 Potsdam, Germany*

**Paul Prasse**                            PAUL.PRASSE@UNI-POTSDAM.DE
*Department of Computer Science, University of Potsdam, 14476 Potsdam, Germany*

**Sebastian Lapuschkin**                   SEBASTIAN.LAPUSCHKIN@HHI.FRAUNHOFER.DE
*Fraunhofer Institute for Telecommunications, Heinrich Hertz Institute, 10587 Berlin, Germany*

**Tobias Scheffer**                        TOBIAS.SCHEFFER@UNI-POTSDAM.DE
*Department of Computer Science, University of Potsdam, 14476 Potsdam, Germany*

**Lena A. Jäger**                          JAEGER@CL.UZH.CH
*Department of Computational Linguistics, University of Zurich, CH-8006 Zürich, Switzerland*

**Editor:** Editor's name

## Abstract

Substantial advances in oculomotoric biometric identification have been made due to deep neural networks processing non-aggregated time series data that replace methods processing theoretically motivated engineered features. However, interpretability of deep neural networks is not trivial and needs to be thoroughly investigated for future eye tracking applications. Especially in medical or legal applications explanations can be required to be provided alongside predictions. In this work, we apply several attribution methods to a state of the art model for eye movement-based biometric identification. To asses the quality of the generated attributions, this work is focused on the quantitative evaluation of a range of established metrics. We find that Layer-wise Relevance Propagation generates the least complex attributions, while DeepLIFT attributions are the most faithful. Due to the absence of a correlation between attributions of these two methods we advocate to consider both methods for their potentially complementary attributions.

## 1. Introduction

Eye movements are known to reflect cognitive processes that include attentional mechanisms (Just and Carpenter, 1976; Henderson, 2003); they are therefore considered to be a *window on the mind and brain* (van Gompel et al., 2007). Eye movements can serve as a basis to automatically screen for ADHD (Deng et al., 2022), dyslexia (Nilsson Benfatto et al., 2016; Raatikainen et al., 2021), and autism spectrum disorder (Alcañiz et al., 2022). Since eye movements are also known to be highly individual (Noton and Stark, 1971), they can be used as a biometric characteristic (Kasprowski and Ober, 2004; Bednarik et al., 2005; Makowski et al., 2021).

---

\* Corresponding Author

For medical screening applications, it is imperative that a machine-learning model can be understood to detect the actual evidence of the condition of interest rather than clever-hans signals or social biases inherent in the training data. Also for biometric identification, it is highly relevant to analyze which signals the model reacts to in order to understand vulnerabilities, biases, and aspects of fairness of the model (Prasse et al., 2022).

In recent years, deep neural networks that process raw gaze-velocity data have achieved dramatic performance increases compared to machine learning on engineered features. For example, recent deep neural network architectures for oculomotoric identification reduces the time for an identification and the error rate by one order of magnitude compared to identification based on engineered saccadic features (Makowski et al., 2021; Lohr and Komogortsev, 2022). Engineered eye-gaze features are often derived from findings from neurophysiological research (Schleicher et al., 2008; Rigas et al., 2018), thus analyses of feature importance of models that rely on such features are meaningful for experts. For deep neural networks, many feature attribution methods have been developed (Bach et al., 2015; Sundararajan et al., 2017; Shrikumar et al., 2017; Smilkov et al., 2017). Unfortunately the interpretability of such complex models is hard and deep neural nets are well known to be black boxes due to the complexity of non-linear activations.

While a wide range of quantitative performance metrics exist for explainability methods, there is no consensus about the merit of each of these metrics. Metrics quantify the complexity (Bhatt et al., 2020; Chalasani et al., 2020; Nguyen and Martínez, 2020), faithfulness (Samek et al., 2015), robustness (Alvarez-Melis and Jaakkola, 2018) or localization (Kohlbrenner et al., 2020; Theiner et al., 2022) of attributions. Image data remains the most popular application domain of these approaches. Unlike gaze data, image data naturally lends itself well to human inspection of attributions, and therefore the evaluation of explainability usually relies in no small part on the plausibility of visualizations of attributions (Adebayo et al., 2018; Bylinskii et al., 2018).

Prior work on applying feature attributions to eye gaze based convolutional neural networks simply quantifies the overall importance of input channels by channel-wise aggregation of feature attributions (Deng et al., 2022). Nevertheless there exists no extensive analysis of feature attributions applied to eye gaze based models up to this date. As a first step towards trusted models and explanations in deep learning based eye gaze applications, in this paper we will therefore rigorously evaluate feature attribution methods with respect to complexity, faithfulness, robustness, and consistency across methods.

In this paper we will restrict ourselves to the task of oculomotoric biometrics, as this is where these models exhibit top performance and it is intuitive that good explanations will also need good predictions (Kindermans et al., 2019). We will train the state-of-the-art neural network *Eye Know You Too* (Lohr and Komogortsev, 2022) on the three publically available datasets *GazeBase* (Griffith et al., 2021), *JuDo1000* (Makowski et al., 2020) and the *Potsdam Textbook Corpus* (Jäger et al., 2021). We will apply a range of feature attribution methods to the trained models, namely *DeepLIFT* (Shrikumar et al., 2017), *Integrated Gradients* (Sundararajan et al., 2017) and *Layer-Wise Relevance Propagation* (Bach et al., 2015), as well as *Input X Gradient* (Shrikumar et al., 2016). We will further quantitatively evaluate these attributions by a variety of established metrics to assess desired properties like complexity, sensitivity, faithfulness and robustness. We will finally evaluate the agreement of attributions across different attribution methods.

The main contributions of this paper are:

- we generate and visualize commonly used attribution methods for a state-of-the-art oculomotoric biometric model,

- we present the first work to evaluate attributions of oculomotoric biometric models on several established metrics and three different real-world datasets,

- we evaluate the agreement between the generated attributions across the attribution methods.

## 2. Materials and Methods

This section is structured as follows: Subsection 2.1 introduces biometric identification and the biometric model under investigation, whereas Subsection 2.2 lists and briefly describes the applied attribution methods. In Subsection 2.3 we give an overview of the employed attribution metrics and in Subsection 2.4 we present the datasets on which the attribution methods are evaluated. Subsection 2.5 lays out the data preprocessing steps and Subsection 2.6 describes the underlying evaluation protocol for each dataset.

### 2.1. Eye Tracking-Based Biometric Identification

We investigate the explainability of a state-of-the-art neural network model for oculomotoric biometric identification, namely *EyeKnowYouToo*, developed by Lohr and Komogortsev (2022). Given a known population of individuals, we investigate a *multi-class classification* setting, where the model is trained using eye movement recording sequences $\{((x_0, y_0) \ldots, (x_n, y_n)\}$ of each user, where $x_i$ and $y_i$ are the yaw (horizontal) and pitch (vertical) gaze eye movement velocities.

*EyeKnowYouToo* (Lohr and Komogortsev, 2022) is a DenseNet-based architecture (Huang et al., 2016) that uses the raw sequences of yaw and pitch angular velocities as input. This end-to-end dilated convolutional network is trained to minimize a multi-similarity loss along with the categorical cross-entropy. We use an extended version of Dillon Lohr's model implementation in PyTorch (Lohr and Komogortsev, 2022).

### 2.2. Attribution Methods

Feature attribution methods are local post-hoc explainability methods that reflect feature importance by attributing positive or negative values to each input feature of a specific model prediction. This facilitates interpretability of given predictions by highlighting relevant parts of the input signal. Attribution methods can be divided into perturbation-based and backpropagation-based methods (Ancona et al., 2017). Due to the computationally expensive approach of perturbation-based methods like SHAP (Lundberg and Lee, 2017) or Occlusion (Zeiler and Fergus, 2014) we limit this study to the evaluation of the backpropagation-based attribution methods Input x Gradient (IxG, Shrikumar et al., 2016), Integrated Gradients (IG, Sundararajan et al., 2017), DeepLIFT (DL, Shrikumar et al., 2017) and Layer-wise relevance propagation (LRP, Bach et al., 2015; Montavon et al., 2017, 2018). Figure 1 presents an example of generated attributions for each of the methods introduced

in this subsection. We use the Captum library (Kokhlikyan et al., 2020) for its IxG, IG and DL implementations and the Zennit library (Anders et al., 2021) for the implementation of LRP rules.

IxG is an early attribution method for which relevance is computed by backpropagating the prediction gradient in respect to each input feature and multiplying the gradient element-wise with the actual input.

LRP computes input relevance by backpropagating the model output to its input according to a specific set of rules. Relevance of each unit is passed down to the lower units depending on the product of activations and weights of the respective layer units and connections, while keeping the total relevance in each layer constant. Although over time a range of different relevance passing rules was proposed (Kohlbrenner et al., 2020), we limit ourselves to the basic original LRP-$\varepsilon$ rule for a more concise presentation. We set $\varepsilon = 0.25std$ according to the parameter selection in Montavon et al. (2019). High $\varepsilon$ values will result in less attributions close to zero, and vice versa for lower $\varepsilon$ values.

DeepLIFT (Shrikumar et al., 2017) is a very similar backpropagation-based attribution method, but in contrast to LRP, an explicit baseline input is used for calculating activation reference points. Activation differences are then backpropagated layer by layer according to a set of rules.

Integrated Gradients (Sundararajan et al., 2017) is somewhat different in that it computes the gradients of the model, which makes it implementation independent. It also uses an explicit baseline, which is then stepwise linearly interpolated into the actual input at hand. For each of those interpolations and for each input feature gradients are calculated, then integrated and finally multiplied with the feature difference between baseline and actual input.

One drawback of attribution methods which use an explicit baseline is the susceptibility for its choice (Sturmfels et al., 2020). Usually a zero or mean baseline is chosen for both of these baseline-based attribution methods, but theoretically every input which leads to a neutral output can be chosen. One drawback of using a constant baseline is the introduced low attribution bias to input values close to the baseline value.

## 2.3. Attribution Metrics

Apart from a qualitative visual analysis we will evaluate the generated attributions by different metrics to measure their complexity, faithfulness and robustness. Due to a lack of ground truth segmentation mask we omit the evaluation of attribution localization. We use the Quantus python package (Hedström et al., 2022) for the calculation of all attribution metrics used in this paper.

Let $\mathbf{x}$ be an instance with $d$ features, $\mathbf{f}$ our model and $\mathbf{g}$ an explanation function, where $\mathbf{g}(\mathbf{f}, \mathbf{x})$ refers to the feature attribution for the model prediction $\mathbf{f}(\mathbf{x})$. We define $\mathbf{g}(\mathbf{f}, \mathbf{x})_i$ as the attribution of the $i$-th input feature. The fractional contribution distribution is defined as $\mathbb{P}_{\mathbf{g}}(i) = \frac{|\mathbf{g}(\mathbf{f}, \mathbf{x})_i|}{\sum_{j \in [d]} |\mathbf{g}(\mathbf{f}, \mathbf{x})_j|}$, for $i \in [d]$, the probability distribution by $\mathbb{P}_{\mathbf{g}} = \{\mathbb{P}_{\mathbf{g}}(i) \mid 0 \le i \le d\}$.

**Complexity metrics** As complexity measures we use entropy (Equation 1, Bhatt et al., 2020), sparseness (Equation 2, Chalasani et al., 2020) and effective complexity (Equation 3, Nguyen and Martínez, 2020) which are defined by the equations below.

The entropy metric is defined as follows:

$$\mu_E(\mathbf{f}, \mathbf{g}, \mathbf{x}) = \mathbb{E}[-\ln(\mathbb{P}_\mathbf{g})] = -\sum_{k=1}^{d} \mathbb{P}_\mathbf{g}(k) \ln(\mathbb{P}_\mathbf{g}(k)) \tag{1}$$

The sparseness metric is based on the Gini coefficient and measures the dispersion between high and low attribution values. It is defined as follows:

$$\mu_S(\mathbf{f}, \mathbf{g}, \mathbf{x}) = 1 - 2\sum_{k=1}^{d} \frac{\mathbf{g}(\mathbf{f}, \mathbf{x})_k}{\|\mathbf{g}(\mathbf{f}, \mathbf{x})\|_1} \frac{d - k + 0.5}{d} \tag{2}$$

Finally the effective complexity measures the amount of absolute attribution values above a certain threshold $\varepsilon$ and is defined as follows:

$$\mu_{EC}(\mathbf{f}, \mathbf{g}, \mathbf{x}) = \frac{1}{d} \left| \left\{ a \in \mathbf{g}(\mathbf{f}, \mathbf{x}) \mid a > \varepsilon \right\} \right| \tag{3}$$

**Faithfulness metric** To measure faithfulness we use the region perturbation metric proposed by (Samek et al., 2015). In this metric we iteratively perturb the input instance $\mathbf{x}$ by non-overlapping patches which are ordered descendently by their sum of inner feature attribution values. The perturbed instance at step $i$ is denoted as $\mathbf{x}_{\text{MoRF}}^{(i)}$.

By iterating through a number of steps $N$ we can thus generate a mean perturbation curve for a set of instances. The underlying intuition is, that perturbing features with high-scoring attributions should lead to a steep drop in target output $\mathbf{f}(\mathbf{x})$ if the evaluated attributions are being actually faithful.

To account for the drop in target output due to model robustness properties, we create a baseline where the non-overlapping order of patches is random. The perturbed instance for step $i$ using this random patch drawing method is denoted as $\mathbf{x}_{\text{Random}}^{(i)}$. The overall metric score is then quantified as the area between the ordered and the random perturbation curve (Equation 4).

$$\mu_{RP}(\mathbf{f}, \mathbf{g}, \mathbf{x}) = \frac{1}{N+1} \sum_{i=1}^{N} \mathbf{f}(\mathbf{x}_{\text{Random}}^{(i)}) - \mathbf{f}(\mathbf{x}_{\text{MoRF}}^{(i)}) \tag{4}$$

**Robustness metrics** As robustness metrics we use the local Lipschitz estimate (Equation 5, Alvarez-Melis and Jaakkola, 2018). This metric perturbs the complete input instance by superimposing noise and measuring the distance between between explanations generated for the perturbed and unperturbed input.

Denote the perturbation of an input instance $\mathbf{x}$ by $\hat{\mathbf{x}}$ and by $\mathcal{N}(\mathbf{x})$ $x$ with added noise drawn from a Gaussian distribution with a mean of 0 and a standard deviation of 0.1. Reusing the notations introduced in the paragraph above, we can define the local Lipshitz estimate as follows:

$$\mu_L(\mathbf{f}, \mathbf{g}, \mathbf{x}) = \arg\max_{\hat{\mathbf{x}} \in \mathcal{N}(\mathbf{x})} \frac{\|\mathbf{g}(\mathbf{f}, \mathbf{x}) - \mathbf{g}(\mathbf{f}, \hat{\mathbf{x}})\|_2}{\|\mathbf{x} - \hat{\mathbf{x}}\|_2} \tag{5}$$

### 2.4. Dataset

We evaluate the model and attributions on the three datasets *GazeBase* (Griffith et al., 2021), *JuDo1000* (Makowski et al., 2020) and the *Potsdam Textbook Corpus (PoTeC)* (Jäger et al., 2021), which are all recorded at a sampling rate of 1000 Hz.

GazeBase is a large scale data set which was gathered over the course of 3 years, consisting of 9 individual rounds with two sessions made on the same day. Although a total number of 322 subjects is available, only 14 subjects participated in the last round. Therefore, we reduce the data set to the first 4 rounds where exactly 100 subjects are available. All subjects participate in seven different tasks: horizontal saccade task (HSS), video viewing task 1 and 2 (VD1 & VD2), a fixation task (FXS), a random saccade task (RAN), a reading task (TEX) and the Balura Game (BLG). We use all available stimuli for evaluation.

JuDo1000 is a single stimulus data set where each trial consists of the sequential presentation of five randomly placed dots. The intervals between each subsequent dot presentation range between 250 ms and 1 s. The dataset consists of 150 subjects recorded in 4 sessions that are at least two weeks apart from each other. During each session the participants are instructed to visually follow five randomly placed dots on the screen. The intervals between each subsequent dot presentation range between 250 ms and 1 s. In contrast to the other two data sets, data is recorded for both eyes (binocular).

PoTeC is a single session data set recorded on a reading task. All 75 participants are instructed to read 12 different short texts.

### 2.5. Data Preprocessing

We base our data preprocessing pipeline on the method proposed by Lohr & Komogortsev (Lohr and Komogortsev, 2022). We first transform positional data into velocity data by applying the Savitzky-Golay differentiation filter (Savitzky and Golay, 1964) with a window size of 7 and an order of 2. We create non-overlapping subsequences with a rolling window approach where we use a window size of 1 second (1000 samples @ 1000 Hz) for *JuDo1000* and *PoTeC*, and a window size of 5 seconds (5000 samples @ 1000 Hz) for *GazeBase*. We exclude all subsequences which need padding or which include more than 50% of missing values and clamp all velocities to $\pm 1000$ °/$s$. Further, we apply z-score normalization and finally replace all missing values with 0. We take use of the pymovements package for preprocessing (Krakowczyk et al., 2022).

### 2.6. Evaluation Protocol

In order to evaluate the introduced attribution methods for this specific task and dataset we apply the following protocols for each datasets: We split the GazeBase dataset by a leave-one-round-out scheme, the JuDo1000 dataset by a leave-one-session-out scheme, and the PoTeC dataset by a leave-one-text-out scheme.

This results in a k-fold cross validation protocol to which we adhere for the complete evaluation pipeline (*GazeBase* and *JuDo1000*: $k = 4$, *PoTeC*: $k = 12$). Each fold includes all data for the respective round/session/text as a test set and the remainder as the training set. Model accuracy as well as attribution metrics are evaluated on the test set of each fold only.

We take the predicted class as the target class to create all attributions. We normalize attribution values by the maximum absolute attribution value of the respective instance.

We evaluate every setting on a AMD EPYC 7742 CPU and a NVIDIA DGX A100 GPU. We train all models using the PyTorch (Paszke et al., 2019) library utilizing the NVIDIA CUDA platform. We implement the model evaluation framework using the scikit-learn (Pedregosa et al., 2011) machine learning package. The code can be found online.[1]

## 3. Results

Literature has shown, that we need high performing model to get reliable explanations (Kindermans et al., 2019). To test the performance of the selected model *Eye Know You Too* (Lohr and Komogortsev, 2022), we followed the evaluation protocol described in Section 2.6, where we train our model to identify different identities in a multi-class setting. From our experiments we can conclude, that our model has a reasonably high mean accuracy of above 90% on all datasets, yielding state-of-the-art performance (see Appendix A for details). The remaining section is structured as follows: Section 3.1 evaluates the different attribution methods in a qualitative manner, where the attribution methods are evaluated in a quantitative manner in Sections 3.2-3.4. In Section 3.5 we evaluate the agreement of the different attribution methods.

### 3.1. Qualitative Attribution Analysis

We start the evaluation of attributions by a qualitative visual analysis to put the subsequent quantitative metrics into perspective. Due to the lack of space we unfortunately have to limit the presentation to a single instance taken from the binocular *JuDo1000* dataset which is presented in Figure 1. In order to get a more complete impression of the generated feature attributions, we refer the interested reader to a selection of additional instances of all three datasets in Appendix B. Apart from the cases where we specifically point to a feature of the given example instance the general observations hold true for the vast amount of the instances across datasets.

We first start with a short inspection of the input signal. There exist absolute velocity peaks at the time steps at roughly 100 ms, 350 ms, 600 ms and 875 ms. Although the peaks are reasonably time-aligned across channels, they are sized different depending on the respective input channel. This stems from differences in yaw and pitch direction of the underlying eye movement event but also from deviations between both eyes. We further observe some low velocity oscillations above the noise floor in the pitch channels of both eyes during the first 100 ms.

When observing the generated attributions we first note the tendency of very high absolute attribution values being time-aligned with the previously identified high velocity eye movements. Still, depending on the attribution method, low attribution values are clearly present in-between these high velocity eye movements. Only LRP exhibits close to zero attribution values at almost all input features not in the temporal vicinity of the identified velocity peaks.

Regarding the consistency of top attribution values among the attribution methods, we observe that methods attribute the pitch velocity of the left eye at about 350 ms (Figure 1(c))

---

1. https://github.com/aeye-lab/2022-nips-gmml-xai-eye-tracking-evaluation

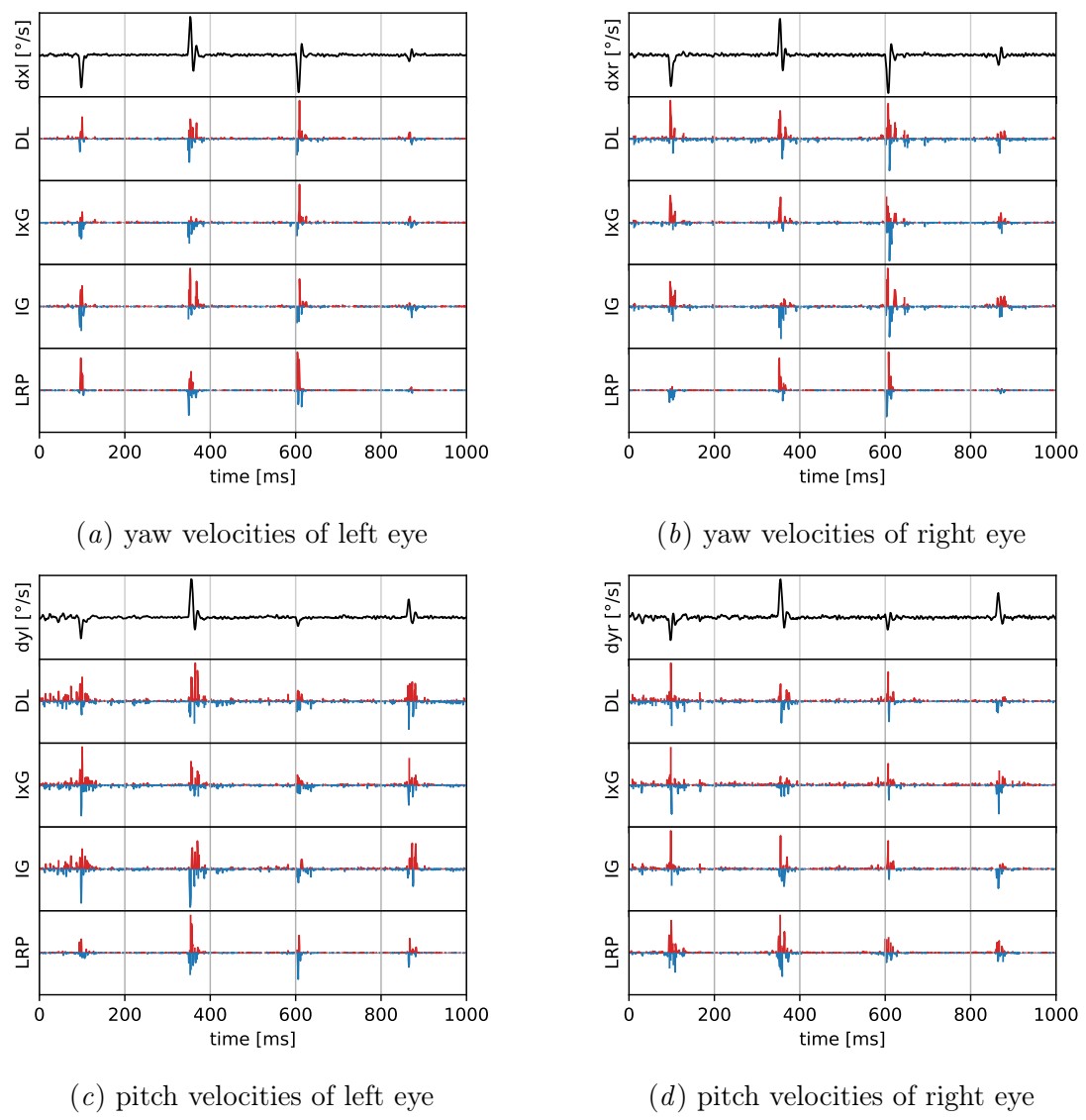

($a$) yaw velocities of left eye          ($b$) yaw velocities of right eye

($c$) pitch velocities of left eye          ($d$) pitch velocities of right eye

Figure 1: Attributions generated by the employed attribution methods (see Subsection 2.2) for a single example instance out of the *JuDo1000* dataset. Each subfigure represents one of the four input velocity channels (from upper left to lower right: (a) yaw left eye, (b) yaw right eye, (c) pitch left eye, (d) pitch right eye). The respective channel signals are plotted as a continuous black line in the first row of each subfigure with y-axis scale from -1000 to 1000 $°/s$. The remaining rows depict the generated feature attributions for the method labeled at the y-axis. Red represents positive attributions and blue represents negative attributions. All attributions are normalized in the range between -1 and 1.

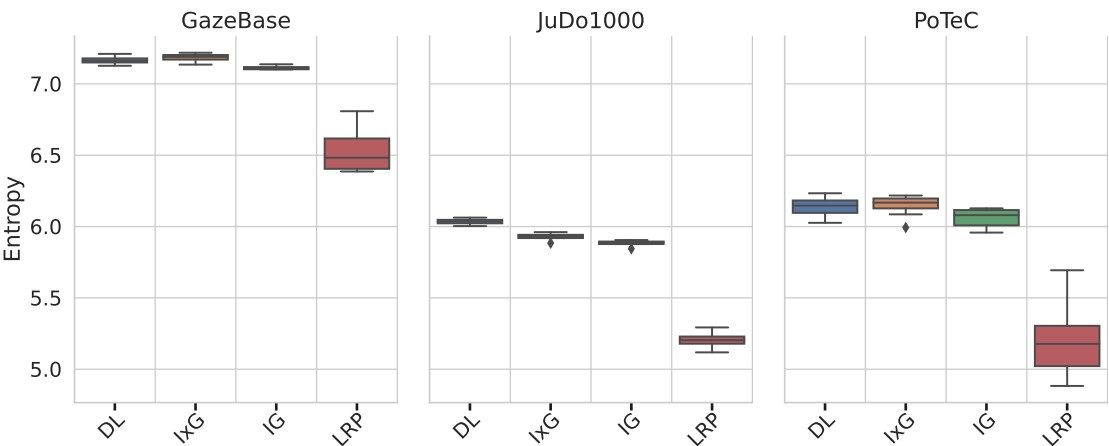

Figure 2: Attribution entropy. The lower the better. See Section 2.3 for metric definitions.

as the most important input feature. Further we see consistent peaks at about 600 ms in both yaw channels (Figure 1(a) and 1(b)).

We observe that LRP doesn't attribute high values to the previously identified low velocity oscillations during the first 100 ms of both pitch velocity channels, while the three other attribution methods attribute much higher values, especially for the left eye (Figure 1(c)).

Another aspect that we note is the seeming ambivalence between positive and negative attributions. Positive attributions express a positive influence of a specific input feature towards the target class, and vice versa with negative attributions. We notice this ambivalence also across channels, for example the IG and LRP attributions during the velocity peak at about 300 ms (left eye pitch velocity in Figure 1(c)) have opposing signs for the rise and fall of the input velocity profile. Taking the mean of attribution values across input channels for each time step would mitigate this issue though, as the positive attributions of each channel outweigh the negative ones at this time step.

### 3.2. Attribution Complexity

We present the results of the quantitative complexity metrics regarding entropy $\mu_E$ in Figure 2, sparseness $\mu_S$ in Figure 3 and effective complexity $\mu_{EC}$ in Figure 4.

For all three complexity metrics we observe that LRP sets itself apart from the other evaluated methods. LRP attributions consistently exhibit less entropy and are more sparse. We further notice distinctively less attribution values between 1e−4 and 0.2 than it is the case for the other methods.

Nevertheless there's a turning point below an $\varepsilon$ of about 1e−4. This shows that attribution values from LRP approach 0 more gradually, while the other methods seem to omit attribution values between 0 and 1e−4. This is just a minor detail, as for a practitioner higher $\varepsilon$ threshold values in the range between 1e−3 and 1e−1 are usually much more relevant.

We observe the exact same rank order for the two best performing methods (LRP and IG) across all three complexity metrics and datasets. There are some inconsistencies across

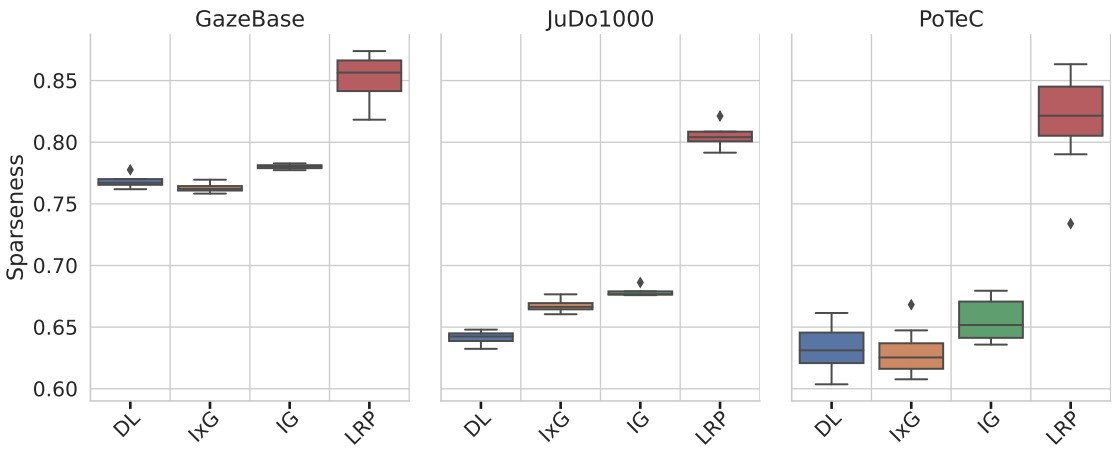

Figure 3: Attribution sparseness. The higher the better. See Section 2.3 for metric definitions.

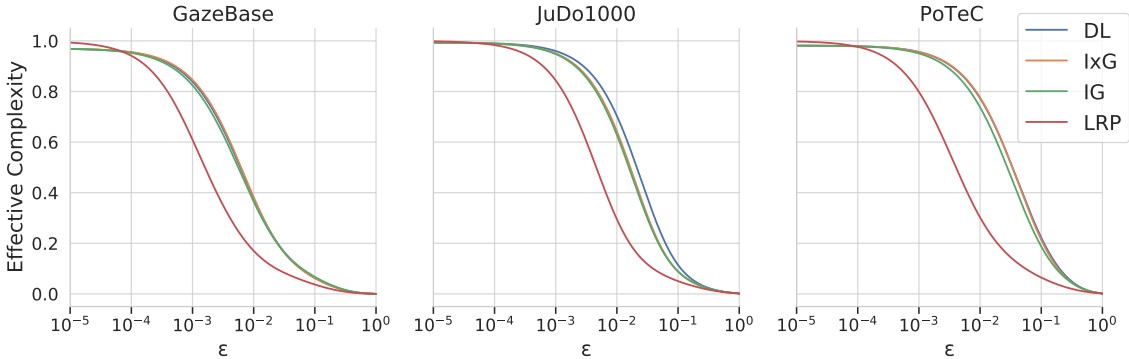

Figure 4: Effective complexity with a logarithmic scale for $\varepsilon$ values. The lower the curve the better. See Section 2.3 for metric definitions.

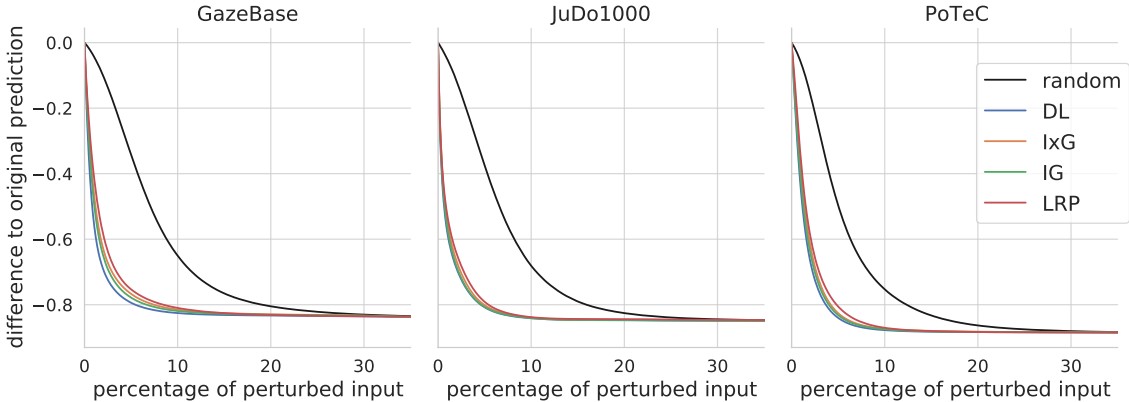

Figure 5: Mean perturbation curves for the employed attribution methods. The mean random perturbation curve is plotted as a continuous black line. The greater the area between the random perturbation curve and the attribution perturbation curve (AOPC relative to random) the better. See Section 2.3 for metric definitions.

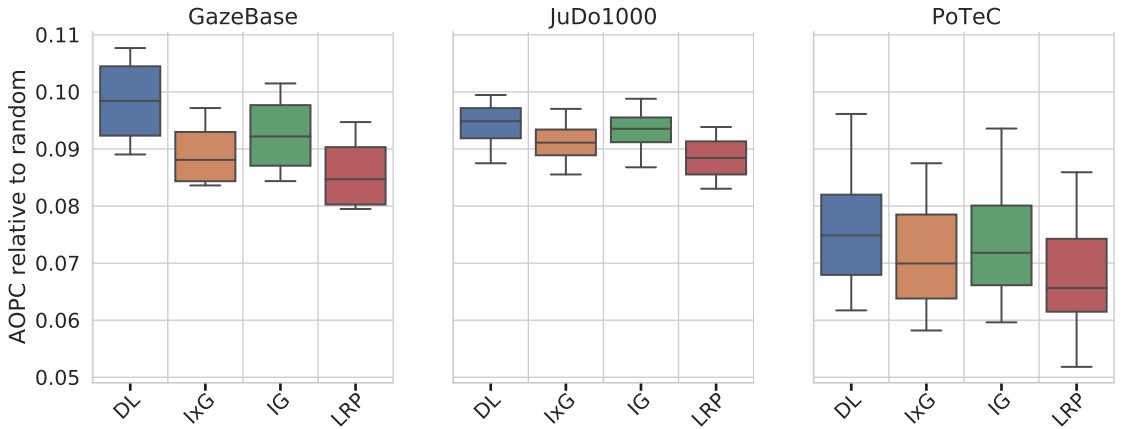

Figure 6: Boxplot for the AOPCs relative to the random perturbation curve ($\mu_{RP}$). The higher the better. See Section 2.3 for metric definitions.

datasets for IxG and DL, with DL performing worse on JuDo1000 and better on GazeBase and PoTeC.

LRP attributions are thus by far the least complex, followed by IG. We note a higher variance for LRP across all datasets, and an overall increase in variance for the PoTeC dataset.

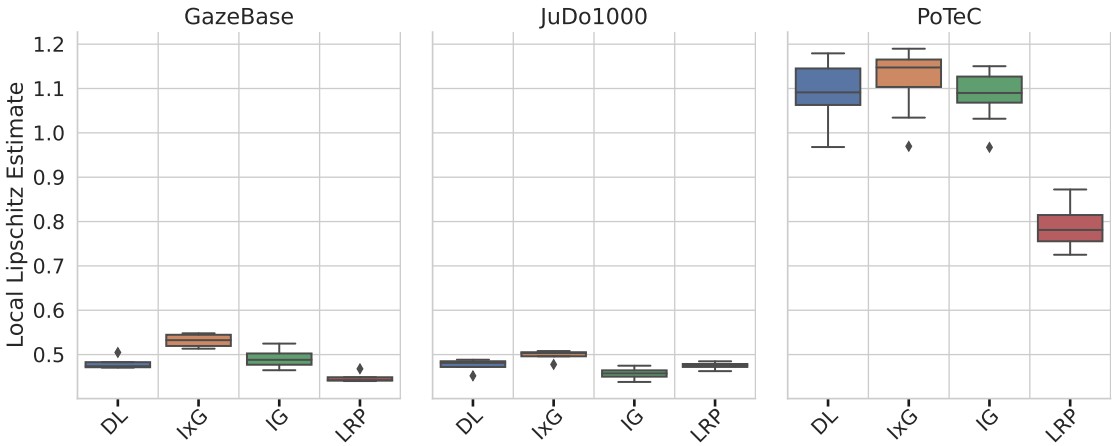

Figure 7: Local Lipschitz estimate. The lower the better. See Section 2.3 for metric definitions.

## 3.3. Attribution Faithfulness

We continue our quantitative attribution evaluation with the faithfulness measure of the region perturbation metric $\mu_{RP}$. As described in the respective metric paragraph in Subsection 2.3 we increasingly perturb the input instance on non-overlapping patches and measure the model output difference for the respective target output.

Figure 5 depicts the mean perturbation curves for each attribution method together with a random perturbation curve. We observe a very similar curve shape across all attribution methods, with LRP being consistently less steep than the other methods. This is in concordance with the boxplot in Figure 6, where we can further see that rank order is preserved across all three datasets. DL and IG are thus the most faithful methods regarding this evaluation. All perturbation curves of each method converge to the random perturbation curve before about 30 % of perturbed input.

However, the boxplot in Figure 6 exposes a relatively high variance for all methods, where mean scores span an interval that is less than 1.5 times the interquartile range. This is especially true for the PoTeC dataset, which exhibits a much bigger variance than the other datasets.

## 3.4. Attribution Robustness

We evaluate the attribution methods on the robustness metric $\mu_L$, which measures the difference in attributions on noise superimposition across the whole input. Figure 7 depicts the respective metric results. We observe several inconsistencies across datasets. Most noticable are the much less robust attributions created on the PoTeC dataset, which potentially stems from the fact that this is a single-session dataset and the trained model is more susceptible to noise superimposition. Unfortunately rank order is not preserved across datasets. The only exception is IxG which performs worst on all three datasets.

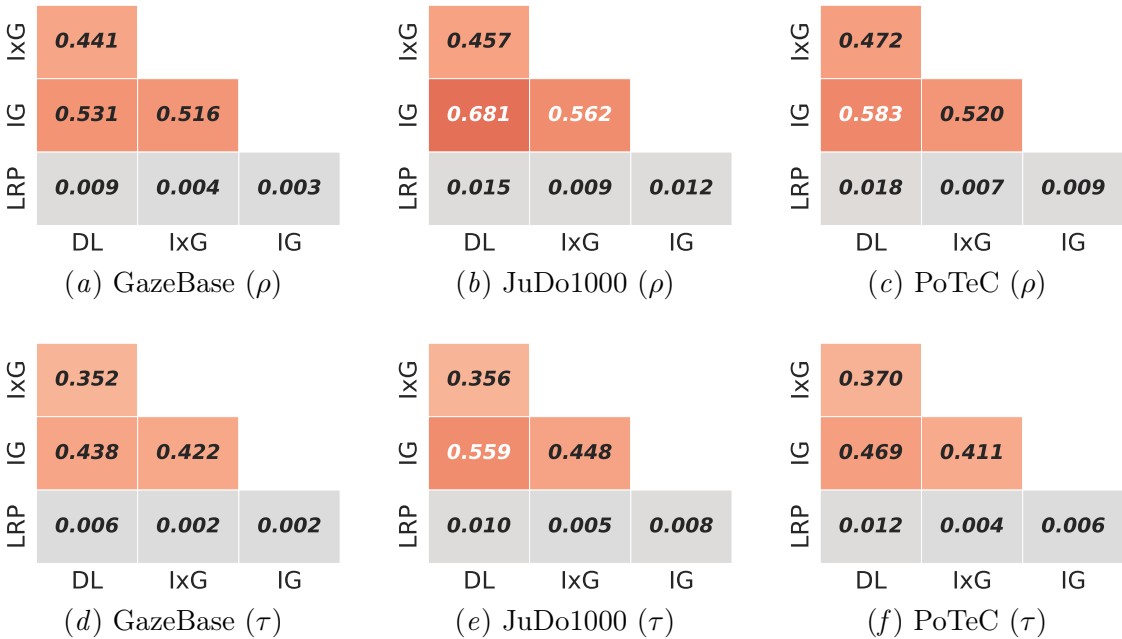

Figure 8: Attribution correlation using Spearman's $\rho$ (a-c) and Kendall's $\tau$ coefficient (d-f). All correlation coefficients are significant with a p-value below $\alpha = 0.001$.

### 3.5. Agreement Across Attribution Methods

We finally evaluate the agreement across attribution methods by correlation analysis using Spearman's $\rho$ coefficient and Kendall's $\tau$ coefficient. The corresponding correlation matrices are depicted in Figure 8.

We note that there is next to no attribution correlation between LRP and the other three methods. The highest correlation is between DL and IG, followed by the correlation between IxG and IG and IxG and DL. Kendall correlations are consistently lower than Spearman correlations while ranks are nevertheless preserved between both correlation methods. The results are consistent across datasets.

## 4. Discussion

We quantitatively evaluated the four attribution methods DeepLIFT (DL), Input x Gradient (IxG), Integrated Gradients (IG) and Layer-wise Relevance Propagation (LRP) for complexity, faithfulness and robustness on three real world datasets and the biometric model *Eye Know You Too* (Lohr and Komogortsev, 2022).

Although we identified LRP to create the least complex attributions, their faithfulness was slightly lacking in relation to the other methods, especially to DL which was the most faithful. This relationship was shown for all three datasets and contributes evidence to suggesting a trade-off between complexity on one hand, and faithfulness on the other. Less complex attributions will probably miss some important relevance relative to more complex

attributions. Nevertheless we can also construct scenarios in which complex attributions include features that were not actually important for the model decision, thus lowering their faithfulness.

Regarding attribution robustness we found low concordance across datasets. Apart from IxG being the worst performing attribution method for this metric, we are not able to generalize these results. We further note that it is imperative to use several sessions for biometric evaluation to diminish the effect of session bias, which can be a plausible reason for the decreased robustness on the *PoTeC* dataset.

The challenge which has to be tackled for each single eye tracking application each time again, is the assessment which of these aspects attract higher priority. This largely depends on the cognitive bias of the recipient of the explanations (Bertrand et al., 2022). Clinical experts will be potentially able to interpret more complex attributions than lay persons.

Moreover it will be interesting to see if we can tune some of these attribution methods in such a way, that an optimal trade-off between certain metrics can be found. Especially LRP with the scalar $\epsilon$ parameter as well as its additional layer rules seems a promising candidate for such an undertaking. Further work will also have to be required in assessing the model influence on the resulting attributions.

We have further shown that LRP has close to no correlation with the other attribution methods when it comes to the two employed rank correlation coefficients Spearman's $\rho$ and Kendall's $\tau$. Although the non-correlation can seem as an issue at first, it can also be beneficial for some applications to have uncorrelated attributions due to the prospect of highlighting features that complement each other. From the correlations we conclude that LRP can complement DL or IG well enough. Due to the slightly higher faithfulness of DL we would advocate for an ensemble of LRP and DL.

Regarding the qualitative analysis of the attribution methods we found a tendency for time-alignment of high velocity peaks and high absolute attribution values for all four attribution methods. This is in concordance with previous literature in which the predictive quality of high velocity eye movement statistics is shown to be high (Holland and Komogortsev, 2011, 2013; Rigas et al., 2016). Nevertheless we can also identify low velocity regions which don't correspond to common eye movement types that still exhibit lower to mid attribution values across DL, IxG and IG. We therefore expect that applying the discussed attribution methods to current and future eye tracking applications will further enhance the still ongoing visual analysis in the field and can potentially help in discovering of new types of eye movement features.

We further identified some limitations of this work. First and most obvious, qualitative visual analysis can only analyze a small subset of the data due to limited human resources. This leaves room for undiscovered phenomena and the criticism of cherry picking example instances. On the one hand qualitative visual analysis by human experts cannot be completely replaced when evaluating attributions, as humans will be the recipients of explanations and there is no way to generally predict human preference on such a broad research problem.

On the other hand, some aspects of the undertaken qualitative analysis can be performed computationally, especially the time-alignment analysis between high absolute velocities and attribution values. Moreover, feature attributions by themselves lack interpretability, especially in tasks where models exhibit better-than-human performance due to the complexity of the input space. As future work we therefore propose to employ eye movement detection

algorithms to quantify attribution localizations in regard to these human interpretable features which can be extracted by computational models.

Last but not least, we visually identified attribution ambivalence for all four attribution methods, where positive and negative attribution values are in close vicinity. This currently leads to issues in interpretability, as these contradictory attributions are hard to interpret. Issues in interpretability of these attributions are especially severe for models with subpar prediction performance, as provision of explanations biases the recipient towards accepting the decision (Jakubik et al., 2022). Further analysis will be needed to correctly assess this issue together with its root cause.

## 5. Conclusion

We have quantitatively evaluated the introduced attribution methods in regard to complexity, faithfulness and robustness on three real world datasets. While Layer-wise relevance propagation exhibits low complexity, attributions generated by DeepLIFT are most faithful. Due to the non-correlation of both methods we advocate for considering both methods for their potentially complementary attributions.

Although we identify similarities across attribution methods through visual analysis and quantitive metrics, we also identify differences and conclude that the selection of the respective attribution method will have decisive influence on derived conclusions.

This work therefore is the starting point and a possible baseline for a line of future research in the eye tracking community. We see future work regarding the tuning of attribution methods and models for achieving better metric results, and improving on human interpretability of attributions through existing eye movement concepts in the psychological literature. We propose that future publications on models in eye tracking research increasingly include measures of explainability to their model evaluation protocols to facilitate and assess the usefulness of the systems in real world problems.

## Acknowledgments

This work was partially funded by the German Federal Ministry of Education and Research (grant 01|S20043).

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

## Appendix A. Model Accuracy

We present a multiclass accuracy boxplot for all three datasets.

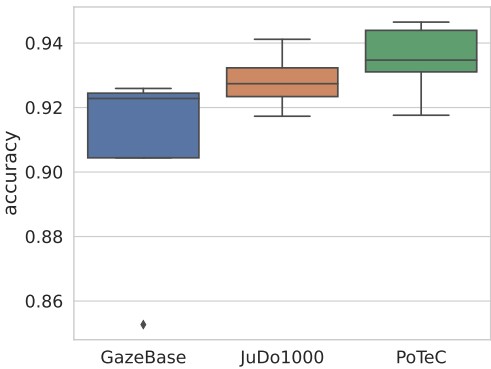

Figure 9: Multiclass accuracy boxplot for the *Eye Know You Too Model* across all three datasets.

# Appendix B. Additional Attribution Examples

This appendix section is dedicated to a brief showcase of the generated attributions for each of the three datasets.

## B.1. Attribution Examples for GazeBase

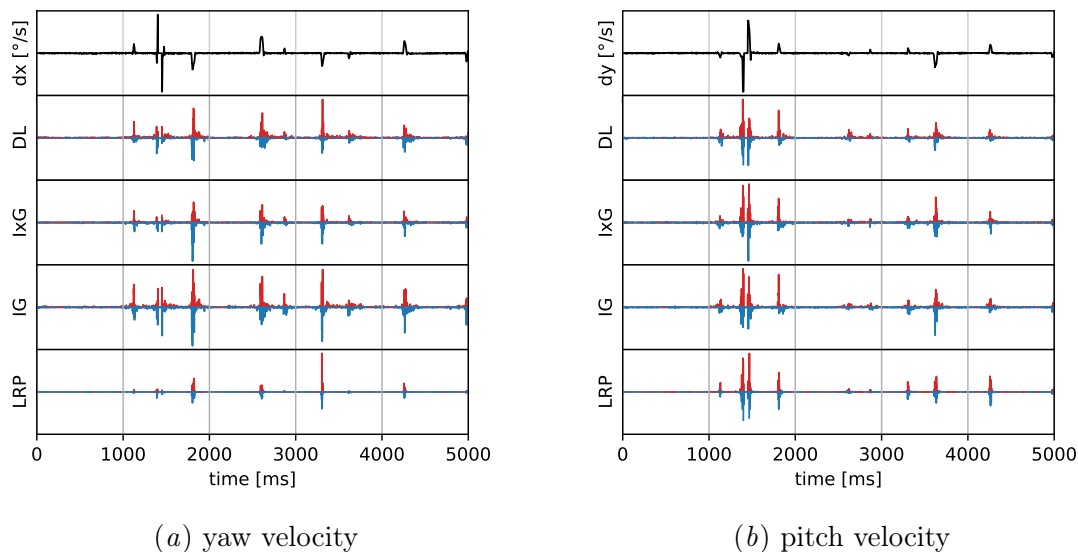

(*a*) yaw velocity          (*b*) pitch velocity

Figure 10: Attributions generated for a single example instance (id = 574) out of the GazeBase dataset. See caption in Figure1 for a complete description.

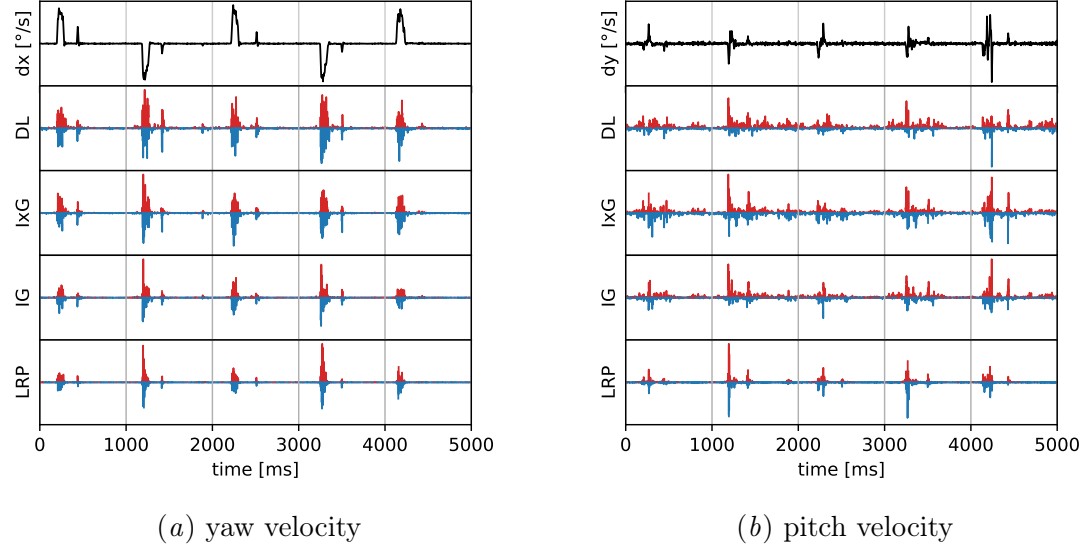

(*a*) yaw velocity                    (*b*) pitch velocity

Figure 11: Attributions generated for a single example instance (id = 2333) out of the GazeBase dataset. See caption in Figure 1 for a complete description.

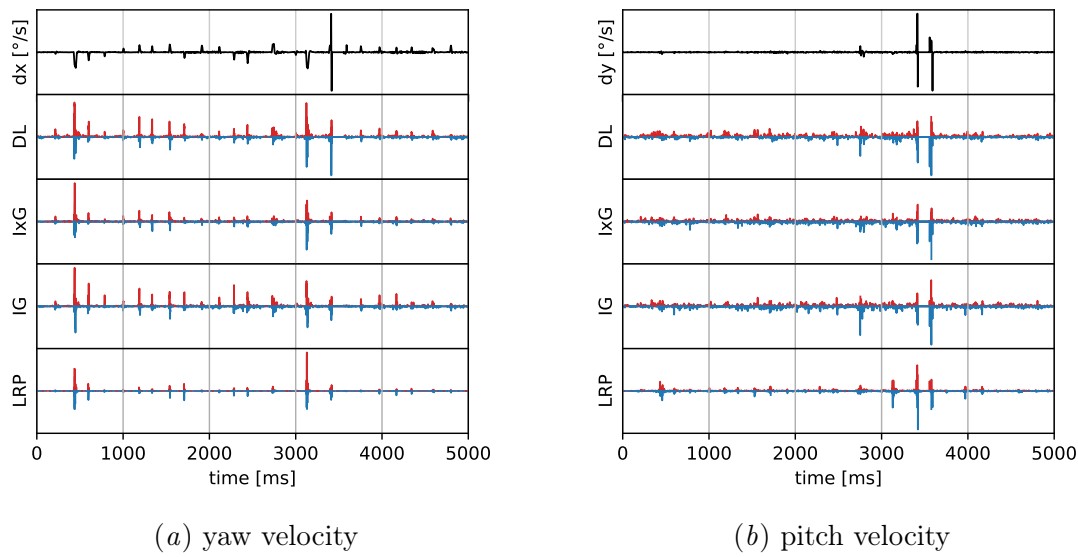

(*a*) yaw velocity                    (*b*) pitch velocity

Figure 12: Attributions generated for a single example instance (id = 13099) out of the GazeBase dataset. See caption in Figure 1 for a complete description.

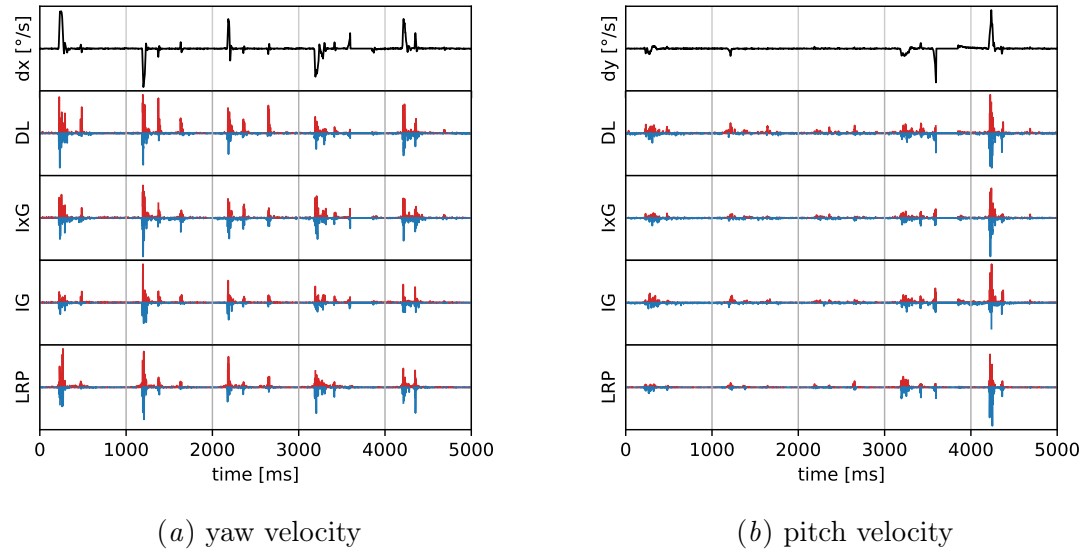

(a) yaw velocity

(b) pitch velocity

Figure 13: Attributions generated for a single example instance (id = 25658) out of the GazeBase dataset. See caption in Figure1 for a complete description.

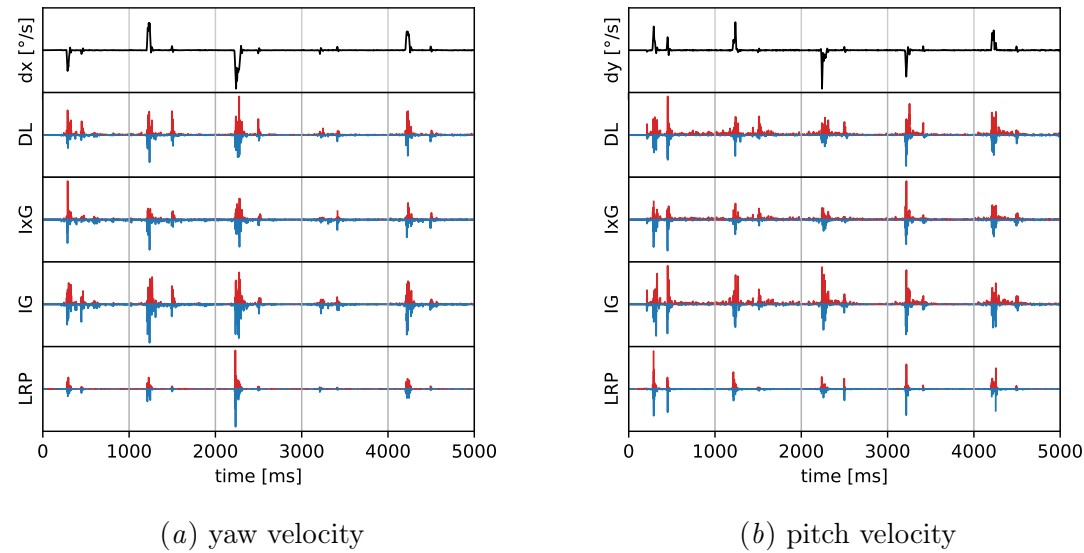

(a) yaw velocity

(b) pitch velocity

Figure 14: Attributions generated for a single example instance (id = 39596) out of the GazeBase dataset. See caption in Figure1 for a complete description.

## B.2. Attribution Examples for JuDo1000

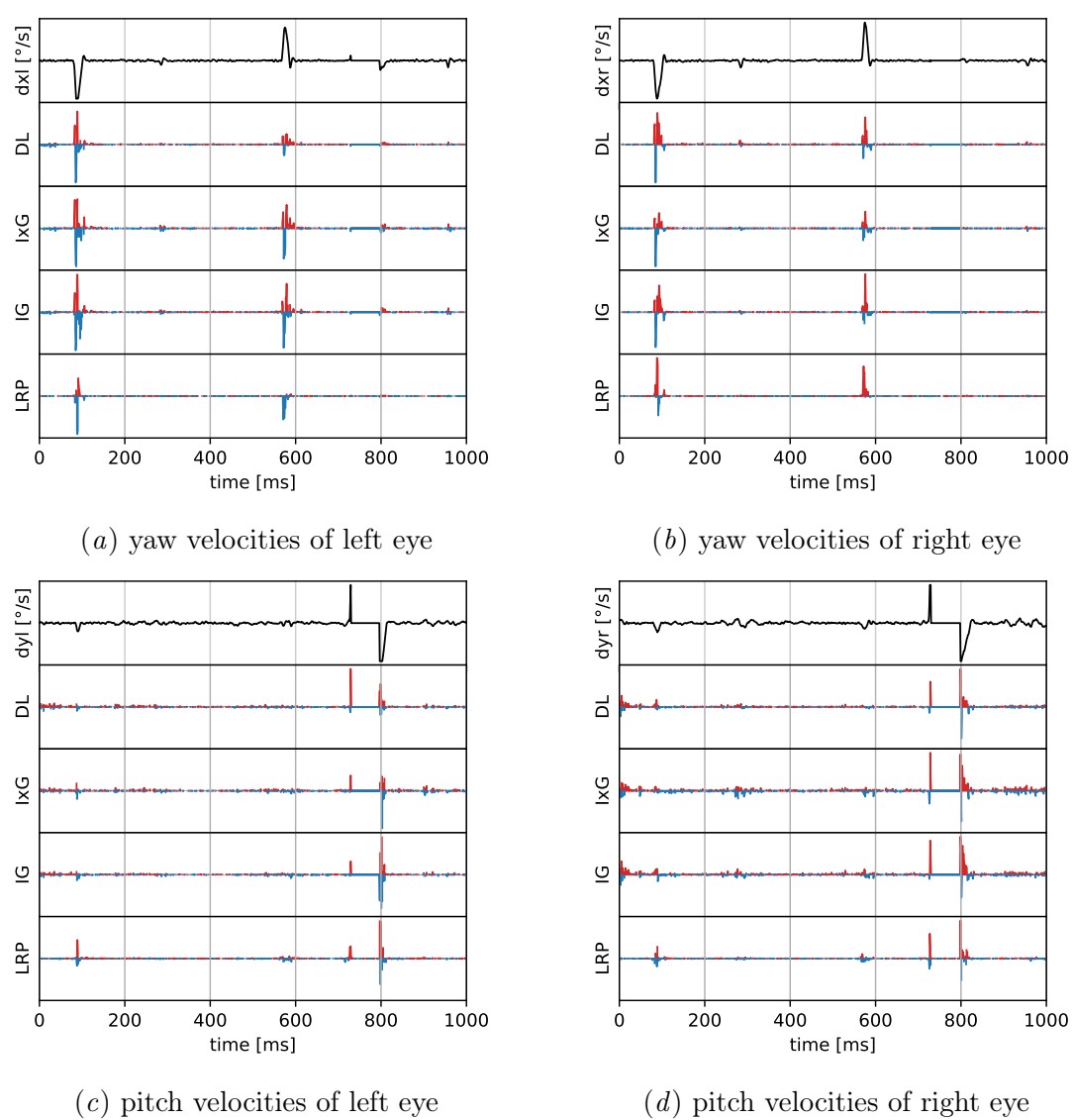

(a) yaw velocities of left eye

(b) yaw velocities of right eye

(c) pitch velocities of left eye

(d) pitch velocities of right eye

Figure 15: Attributions generated for a single example instance (id = 156) out of the JuDo1000 dataset. See caption in Figure1 for a complete description.

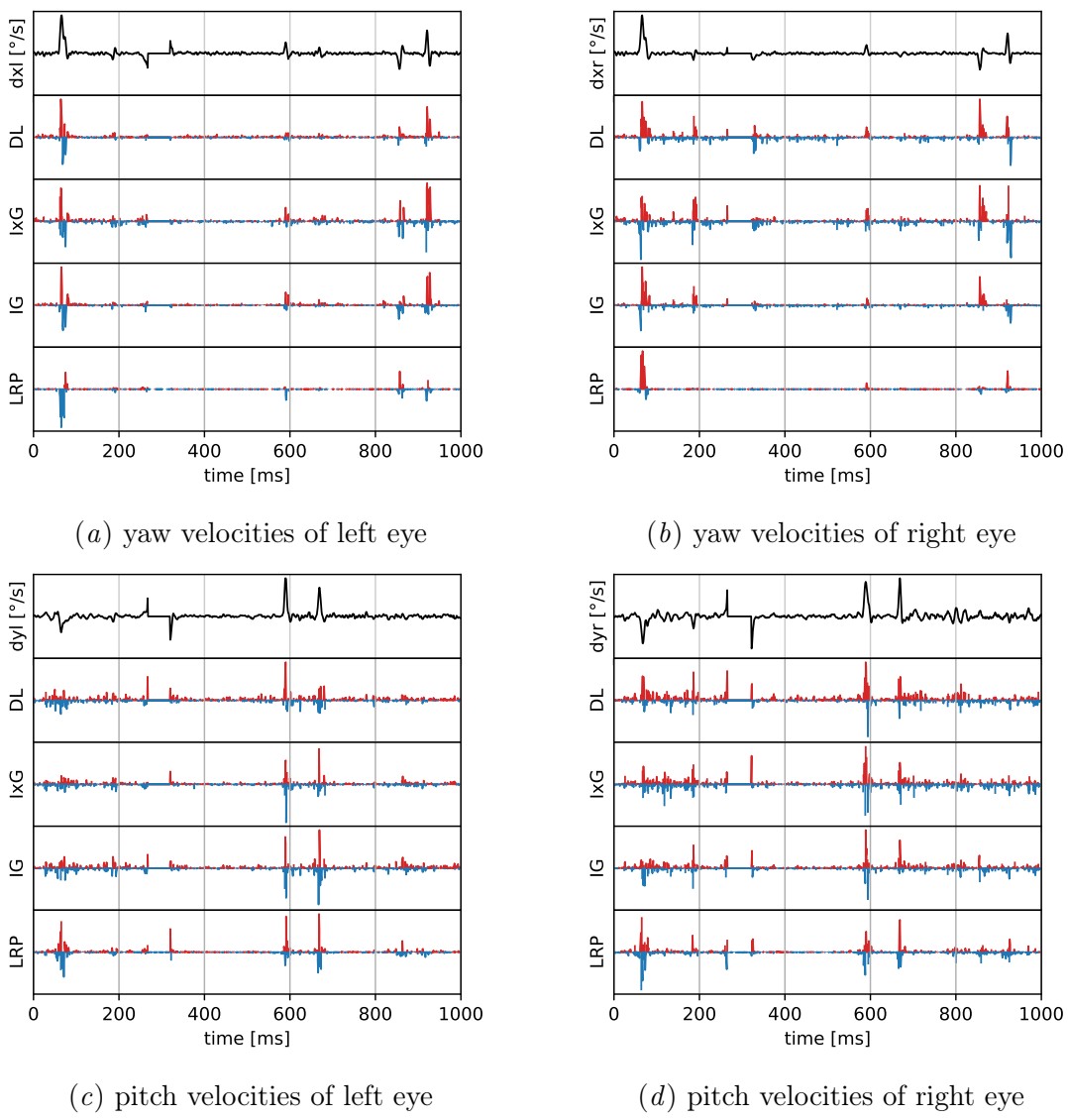

(a) yaw velocities of left eye

(b) yaw velocities of right eye

(c) pitch velocities of left eye

(d) pitch velocities of right eye

Figure 16: Attributions generated for a single example instance (id = 1578) out of the JuDo1000 dataset. See caption in Figure 1 for a complete description.

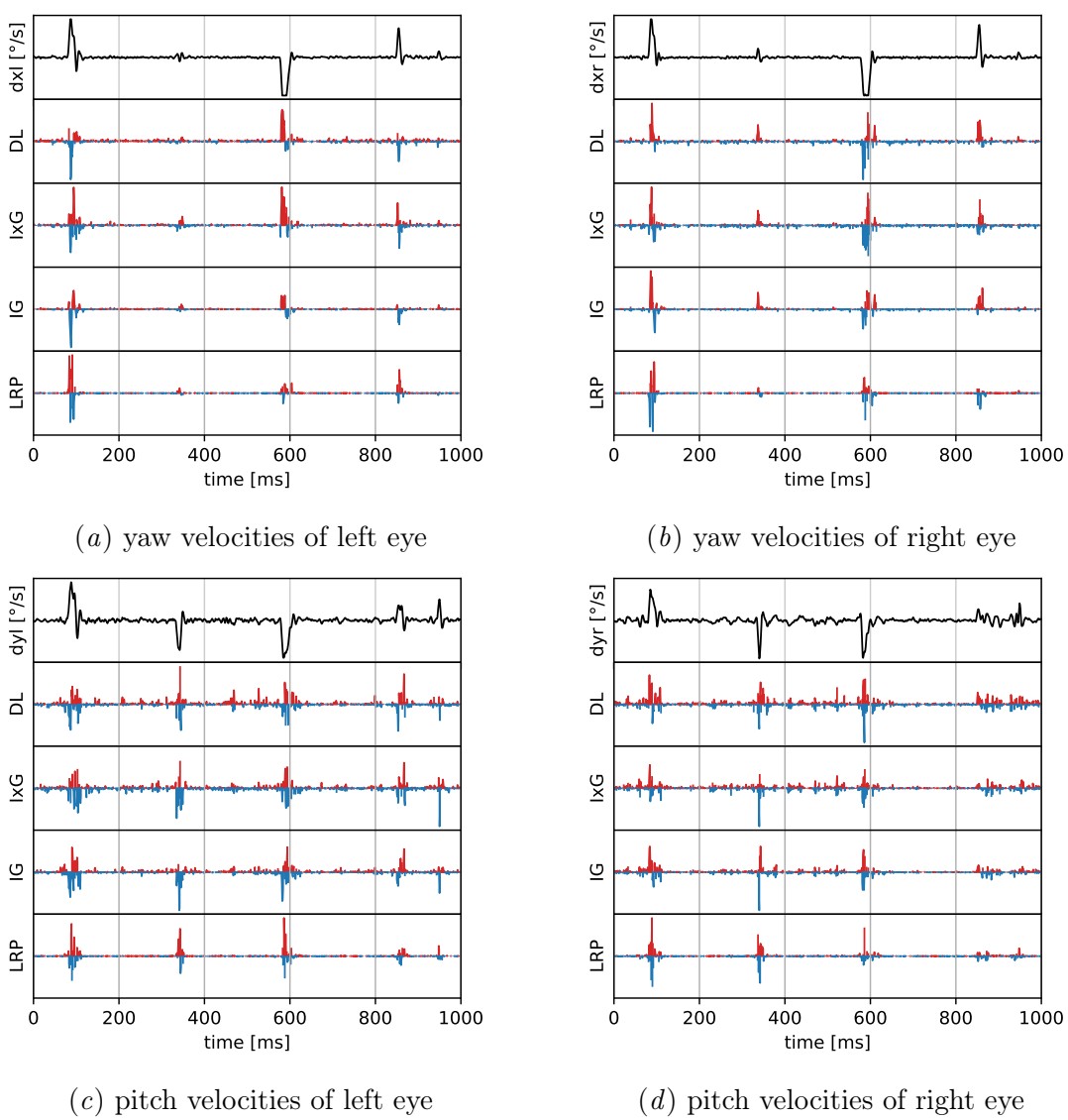

(a) yaw velocities of left eye

(b) yaw velocities of right eye

(c) pitch velocities of left eye

(d) pitch velocities of right eye

Figure 17: Attributions generated for a single example instance (id = 6797) out of the JuDo1000 dataset. See caption in Figure 1 for a complete description.

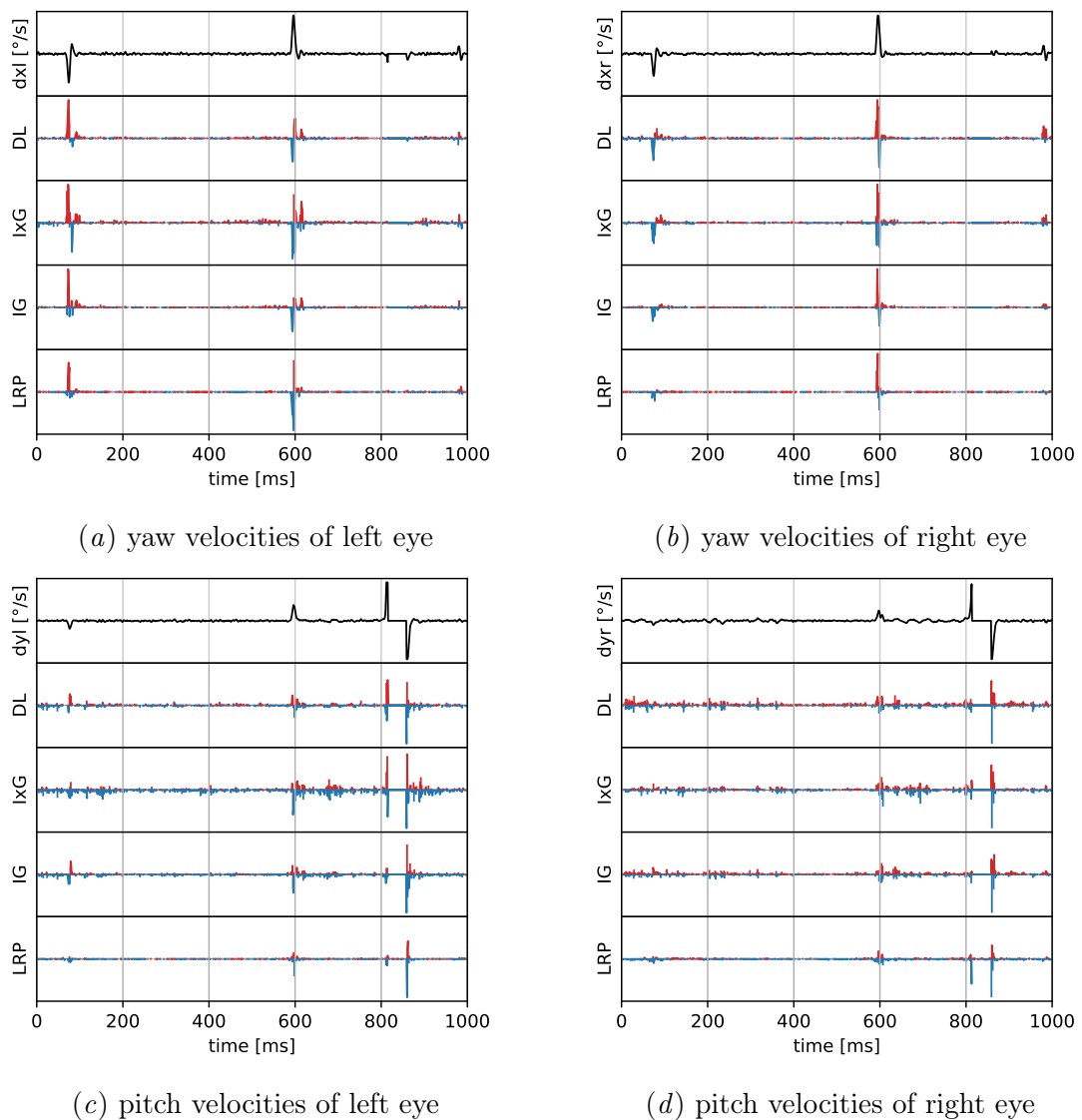

(a) yaw velocities of left eye

(b) yaw velocities of right eye

(c) pitch velocities of left eye

(d) pitch velocities of right eye

Figure 18: Attributions generated for a single example instance (id = 16066) out of the JuDo1000 dataset. See caption in Figure1 for a complete description.

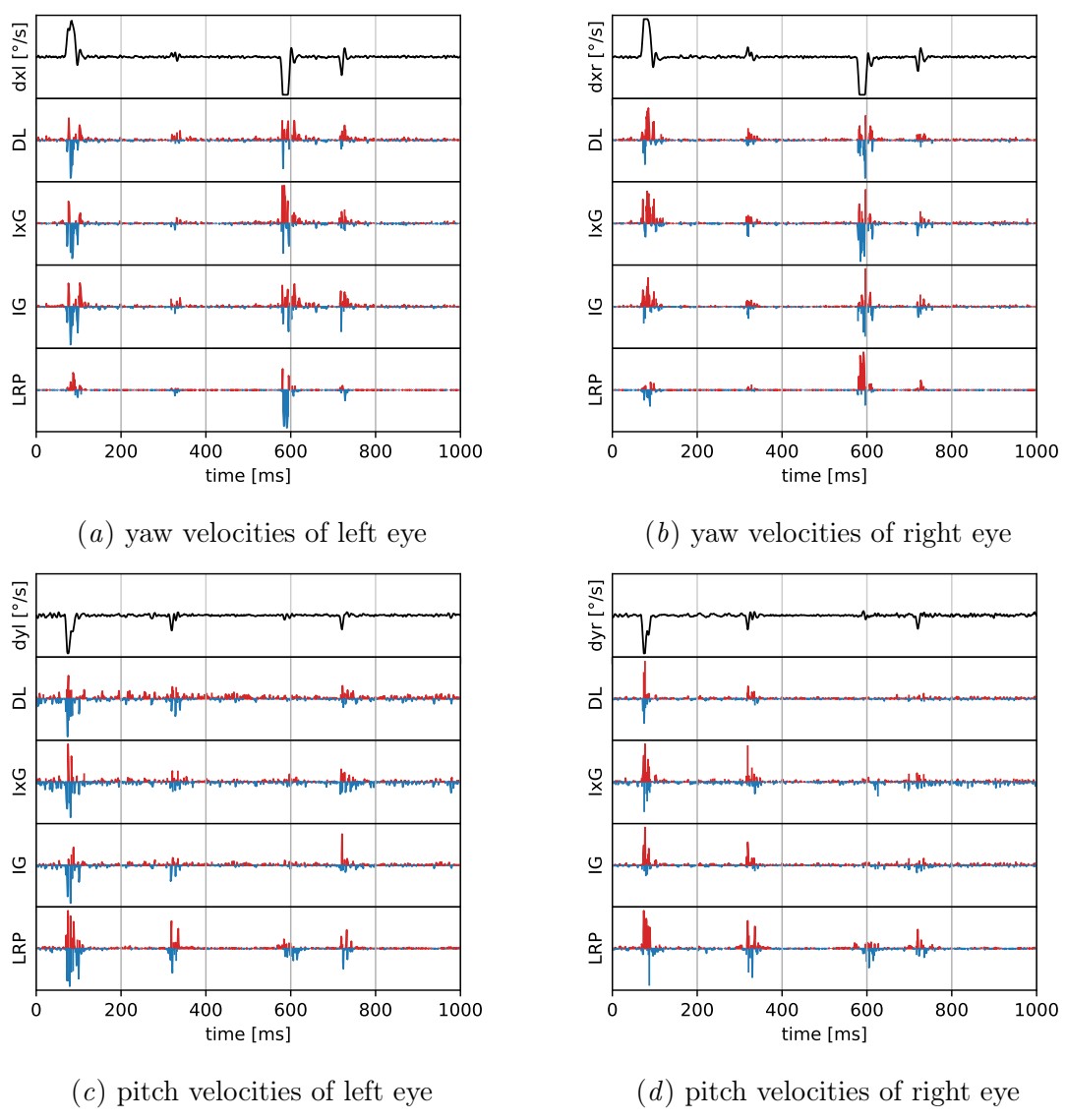

(a) yaw velocities of left eye

(b) yaw velocities of right eye

(c) pitch velocities of left eye

(d) pitch velocities of right eye

Figure 19: Attributions generated for a single example instance (id = 19239) out of the JuDo1000 dataset. See caption in Figure 1 for a complete description.

### B.3. Attribution Examples for Potsdam Textbook Corpus

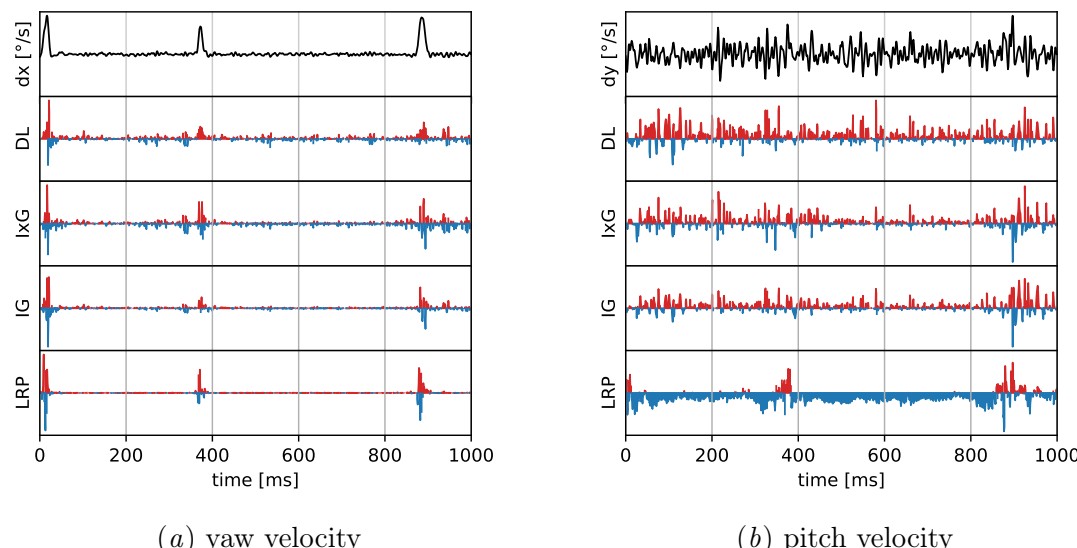

($a$) yaw velocity                    ($b$) pitch velocity

Figure 20: Attributions generated for a single example instance (id = 2164) out of the PoTeC dataset. See caption in Figure 1 for a complete description.

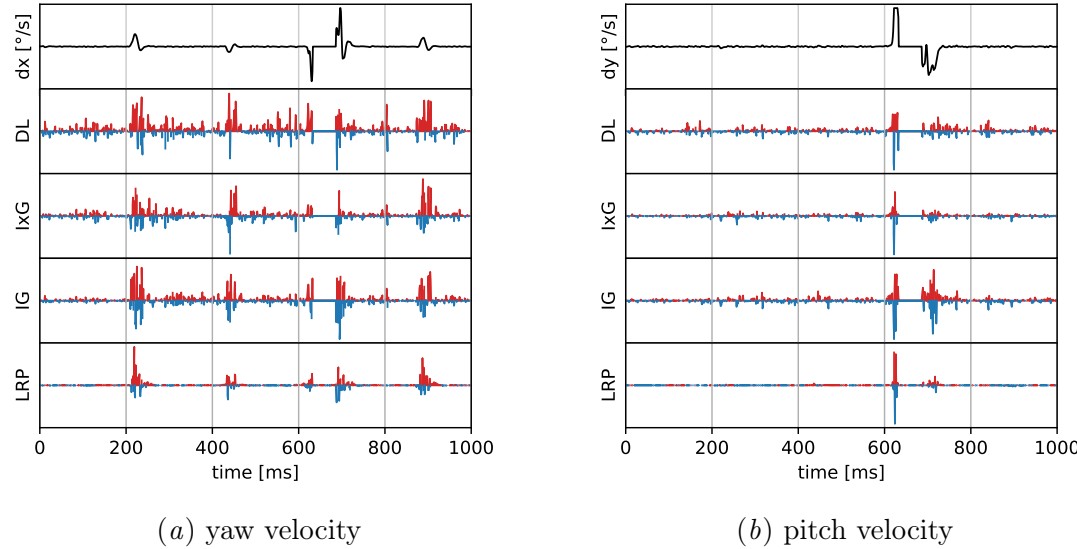

(*a*) yaw velocity          (*b*) pitch velocity

Figure 21: Attributions generated for a single example instance (id = 16137) out of the PoTeC dataset. See caption in Figure 1 for a complete description.

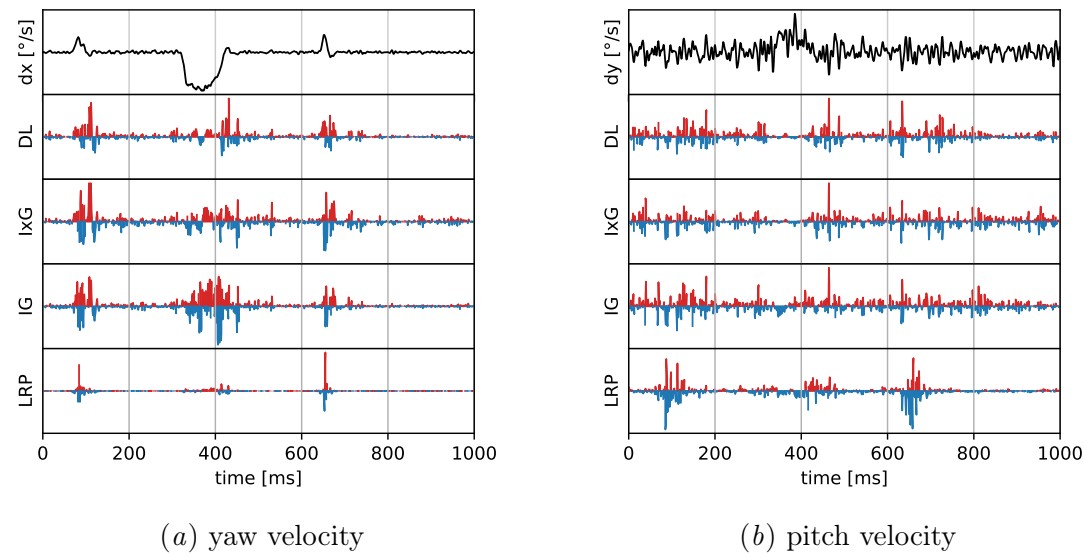

(*a*) yaw velocity          (*b*) pitch velocity

Figure 22: Attributions generated for a single example instance (id = 19806) out of the PoTeC dataset. See caption in Figure 1 for a complete description.

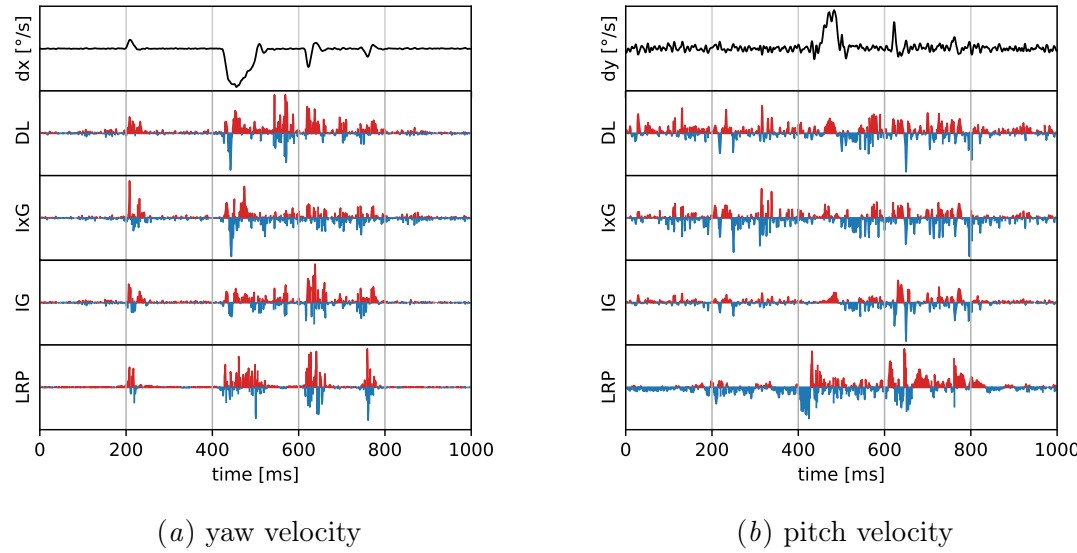

(*a*) yaw velocity  (*b*) pitch velocity

Figure 23: Attributions generated for a single example instance (id = 82639) out of the PoTeC dataset. See caption in Figure 1 for a complete description.

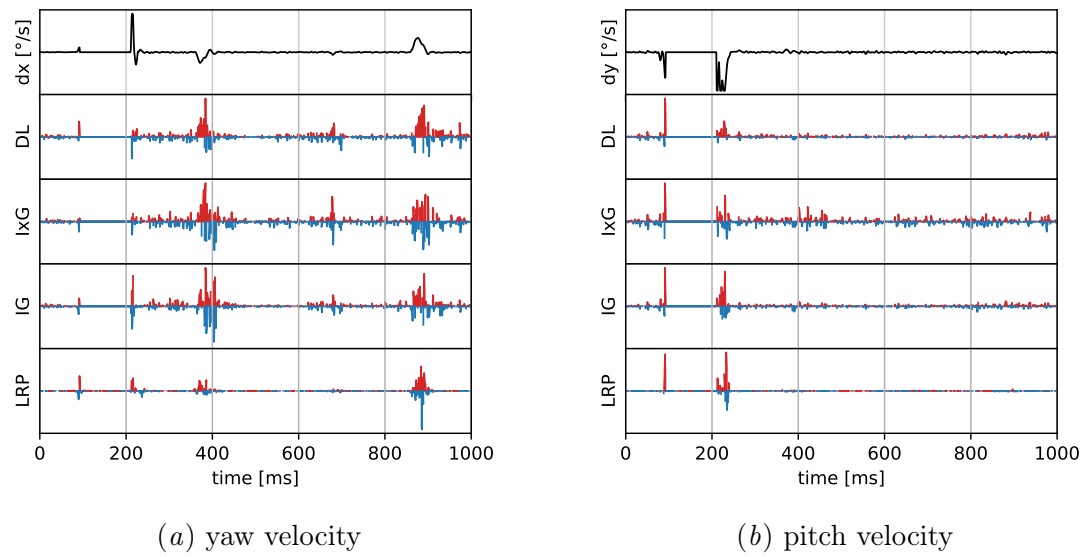

(*a*) yaw velocity  (*b*) pitch velocity

Figure 24: Attributions generated for a single example instance (id = 83911) out of the PoTeC dataset. See caption in Figure 1 for a complete description.

