# OpenReview forum: "Selection of XAI Methods Matters: Evaluation of Feature Attribution Methods for Oculomotoric Biometric Identification"
_NeurIPS.cc/2022/Workshop/GMML — Gaze Meets ML 2022 Poster_

### Official Review · Reviewer_rCE2 · 2022-10-16
**Attribution analysis of eye gaze tracking models with input data**

**Rating:** 8
**Confidence:** 3

**Review:**

This is a very well designed study. Despite the fact that the authors are not proposing major new methodology, their thoughtful methodology of ranking attribution methods is valuable. They have a clear experimental set up, detailed results and conclusions.

Suggestion: please change the title and remove the abbreviation XIA

---

### Official Review · Reviewer_4Fuq · 2022-10-20
**Quantitative evaluation of a single existing method on a single existing dataset leveraging existing metrics.**

**Rating:** 5
**Confidence:** 3

**Review:**

In this work the authors use a single biometric model (EyeKnowYouToo) on an existing public dataset, to quantitatively evaluate four existing backpropagation-based attribution methods for complexity, robustness, and faithfulness, leveraging existing well-defined metrics.


QUALITY & CLARITY

The evaluation is conducted in a thoughtful but also expected manner.

The authors make hypothetical assertions in the discussion section, without any citations to support them, and hence rendered unjustified.

The writing style and language used in the paper are exceptional. In fact, the current way that the paper is written should make its complete scope and approach explicable to even a reader out of the domain.


ORIGINALITY

Lack of novelty is the main disadvantage of this paper.

I would expect to see the proposition of a novel way/measure to quantify these existing methods.


SIGNIFICANCE

Comparative evaluation and descriptive analysis of perturbation-based methods, would strengthen this paper.
Inclusion of another method and dataset, rather than only the EyeKnowYouToo method and the JuDo1000, would show if the current findings are reproducible to the problem or specific to this method and dataset.

---

### Meta-Review · Area_Chair_k5mT · 2022-10-20

**Recommendation:** Accept (Poster)
**Confidence:** 4

**Metareview:**

The authors propose to track the oculomotoric biometric identification using an EyeKnowYouToo model on the JuDo1000 dataset. Then, using available metrics, they present a quantitative evaluation of existing attribution methods for complexity, faithfulness, and robustness.

All the reviewers appreciated the clear presentation and experiments in the paper. However, all the reviewers mentioned the lack of new methodology or innovation. But overall, I think this is an interesting piece of the contribution that can be discussed in a workshop setting.

Overall I recommend an acceptance

---

### Decision · Program_Chairs · 2022-10-20

Accept (Poster)